# LISA: Reasoning Segmentation via Large Language Model

## Abstract

Although perception systems have made remarkable advancements in recent years, they still rely on explicit human instruction to identify the target objects or categories before executing visual recognition tasks. Such systems lack the ability to actively reason and comprehend implicit user intentions. In this work, we propose a new segmentation task — *reasoning segmentation*. The task is designed to output a segmentation mask given a complex and implicit query text. Furthermore, we establish a benchmark comprising over one thousand image-instruction pairs, incorporating intricate reasoning and world knowledge for evaluation purposes. Finally, we present LISA: large **L**anguage **I**nstructed **S**egmentation **A**ssistant, which inherits the language generation capabilities of the multi-modal Large Language Model (LLM) while also possessing the ability to produce segmentation masks. We expand the original vocabulary with a `<SEG>` token and propose the embedding-as-mask paradigm to unlock the segmentation capability. Remarkably, LISA can handle cases involving: 1) complex reasoning; 2) world knowledge; 3) explanatory answers; 4) multi-turn conversation. Also, it demonstrates robust zero-shot capability when trained exclusively on reasoning-free datasets. In addition, fine-tuning the model with merely 239 reasoning segmentation image-instruction pairs results in further performance enhancement. Experiments show our method not only unlocks new reasoning segmentation capabilities but also proves effective in both complex reasoning segmentation and standard referring segmentation tasks.

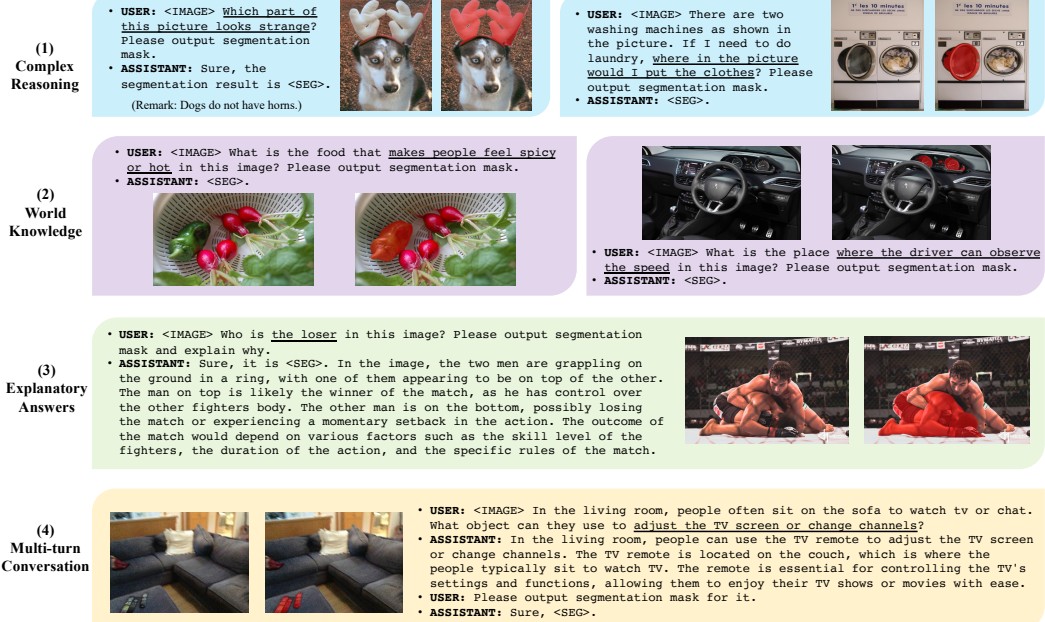

Figure 1: We unlock new segmentation capabilities for current multi-modal LLMs. The resulting model (LISA) is capable to deal with cases involving: (1) Complex Reasoning; (2) World Knowledge; (3) Explanatory Answers; (4) Multi-turn Conversation.

# 1 INTRODUCTION

In daily life, users tend to issue direct commands like "Change the TV channel" to instruct a robot, rather than providing explicit step-by-step instructions such as "Go to the table first, find the TV remote, and then press the button to change the channel." However, existing perception systems consistently rely on humans to explicitly indicate target objects or pre-define categories before executing visual recognition tasks. These systems lack the capacity to actively reason and comprehend users' intentions based on implicit instructions. This reasoning ability is crucial in developing next-generation intelligent perception systems and holds substantial potential for industrial applications, particularly in robotics.

In this work, we introduce a new segmentation task — *reasoning segmentation*, which requires generating a binary segmentation mask based on an implicit query text involving *complex reasoning*. Notably, the query text is not limited to a straightforward reference (e.g., "the orange"), but a more complicated description involving *complex reasoning* or *world knowledge* (e.g., "the food with high Vitamin C"). To accomplish this task, the model must possess two key abilities: 1) reasoning *complex* and *implicit* text queries jointly with the image; 2) producing segmentation masks.

Inspired by the exceptional capacity of the Large Language Model (LLM) to reason and comprehend user intentions, we aim to leverage this capability to address the aforementioned first challenge. However, while several studies have integrated robust reasoning capabilities into multi-modal LLMs to accommodate visual input, the majority of these models primarily concentrate on text generation tasks and still fall short in performing vision-centric tasks that necessitate fine-grained output formats, such as segmentation masks.

In this work, we introduce LISA: a large **L**anguage **I**nstructed **S**egmentation **A**ssistant, a multi-modal LLM capable of producing segmentation masks. To equip the multi-modal LLM with segmentation abilities, we incorporate an additional token, i.e., `<SEG>`, into the existing vocabulary. Upon generating the `<SEG>` token, its hidden embedding is further decoded into the corresponding segmentation mask. By representing the segmentation mask as an embedding, LISA acquires segmentation capabilities and benefits from end-to-end training. Remarkably, LISA demonstrates robust zero-shot abilities. Training the model solely on standard semantic segmentation and referring segmentation datasets yields surprisingly effective performance on the complex reasoning segmentation task. Furthermore, we find that LISA's performance can be significantly enhanced by fine-tuning on just 239 image-instruction reasoning segmentation pairs. As illustrated in Fig. 1, LISA can handle various scenarios, including: 1) complex reasoning; 2) world knowledge; 3) explanatory answers; and 4) multi-turn conversations.

In addition, to validate the effectiveness, we establish a benchmark for reasoning segmentation evaluation, called *ReasonSeg*. Comprising over one thousand image-instruction pairs, this benchmark offers persuasive evaluation metrics for the task. To align more closely with practical applications, we annotate the images from OpenImages (Kuznetsova et al., 2020) and ScanNetv2 (Dai et al., 2017) with implicit text queries that necessitate complex reasoning.

In summary, our contributions are as follows:

- We introduce the *reasoning segmentation* task, which necessitates reasoning based on implicit human instructions. This task emphasizes the importance of self-reasoning ability, crucial for building a genuinely intelligent perception system.

- We establish a reasoning segmentation benchmark, *ReasonSeg*, containing over one thousand image-instruction pairs. This benchmark is essential for evaluation and encourages the community to develop new techniques.

- We present our model — LISA, which employs the embedding-as-mask paradigm to incorporate new segmentation capabilities. LISA demonstrates robust zero-shot ability on the reasoning segmentation task when trained on reasoning-free datasets and achieves further performance boost by fine-tuning on just 239 image-instruction pairs involving reasoning. We believe LISA will promote the development of perceptual intelligence and inspire new advancements in this direction.

## 2 RELATED WORK

### 2.1 IMAGE SEGMENTATION

Semantic segmentation aims to assign a class label to every pixel in an image. Numerous studies (Shelhamer et al., 2017; Noh et al., 2015; Badrinarayanan et al., 2017; Ronneberger et al., 2015; Chen et al., 2018; Yu & Koltun, 2016; Liu et al., 2015; Zhao et al., 2017; 2018a; Yang et al., 2018; Fu et al., 2019; Huang et al., 2019; Zhao et al., 2018b; Zhu et al., 2019; Cheng et al., 2021; Lai et al., 2021; Tian et al., 2022; 2023) have proposed diverse designs (such as encoder-decoder, dilated convolution, pyramid pooling module, non-local operator, and more) to effectively encode semantic information. Research on instance segmentation (He et al., 2017; Zhang et al., 2021; Cheng et al., 2022) and panoptic segmentation (Kirillov et al., 2019; Xiong et al., 2019; Cheng et al., 2020; Li et al., 2021) has introduced various architectural innovations for instance-level segmentation, including DETR (Carion et al., 2020)-based structures, mask attention, and dynamic convolution. In recent years, typical segmentation tasks have made significant progress and become increasingly mature. Consequently, it is imperative to develop more intelligent interaction ways for image segmentation.

The referring segmentation task (Kazemzadeh et al., 2014; Nagaraja et al., 2016) enables interaction with human language, aiming to segment the target object based on a given explicit text description. Recently, Kirillov et al. (2023) introduced SAM, trained with billions of high-quality masks, supporting bounding boxes and points as prompts while demonstrating exceptional segmentation quality. X-Decoder (Zou et al., 2023a) bridges vision and language, unifying multiple tasks within a single model. SEEM (Zou et al., 2023b) further supports various human interaction methods, including text, audio, and scribble. However, these studies primarily focus on addressing multi-task compatibility and unification, neglecting the injection of new capabilities. In this work, we present LISA to tackle the reasoning segmentation task and enhance existing visual segmentors with self-reasoning abilities.

### 2.2 MULTI-MODAL LARGE LANGUAGE MODEL

Motivated by the remarkable reasoning abilities of LLMs, researchers are exploring ways to transfer these capabilities into the vision domain, developing multi-modal LLMs. Flamingo (Alayrac et al., 2022) employs a cross-attention structure to attend to visual contexts, enabling visual in-context learning. Models such as BLIP-2 (Li et al., 2023b) and mPLUG-OWL (Ye et al., 2023) propose encoding image features with a visual encoder, which are then fed into the LLM alongside text embeddings. Otter (Li et al., 2023a) further incorporates robust few-shot capabilities through in-context instruction tuning on the proposed MIMIC-IT dataset. LLaVA (Liu et al., 2023b) and MiniGPT-4 (Zhu et al., 2023) first conduct image-text feature alignment followed by instruction tuning. Koh et al. (2023) also investigates image retrieval for LLMs. Moreover, numerous works (Wu et al., 2023; Yang et al., 2023b; Shen et al., 2023; Liu et al., 2023c; Yang et al., 2023a) utilize prompt engineering, connecting independent modules via API calls, but without the benefits of end-to-end training. Recently, there have been studies examining the intersection between multi-modal LLMs and vision tasks. VisionLLM (Wang et al., 2023) offers a flexible interaction interface for multiple vision-centric tasks through instruction tuning but fails to fully exploit LLMs for complex reasoning. Kosmos-2 (Peng et al., 2023) constructs large-scale data of grounded image-text pairs, infusing grounding capabilities into LLMs. DetGPT (Pi et al., 2023) bridges the fixed multi-modal LLM and open-vocabulary detector, enabling detection to be performed based on users' instructions. GPT4RoI (Zhang et al., 2023a) introduces spatial boxes as input and trains the model on region-text pairs. In contrast, our work aims to 1) efficiently inject segmentation capabilities into multi-modal LLMs and 2) unlock self-reasoning abilities for current perception systems.

## 3 REASONING SEGMENTATION

### 3.1 PROBLEM DEFINITION

The reasoning segmentation task is to output a binary segmentation mask $M$, given an input image $x_{img}$ and an implicit query text instruction $x_{txt}$. The task shares a similar formulation with the referring segmentation task (Kazemzadeh et al., 2014), but is far more challenging. The key distinction lies in the complexity of the query text in reasoning segmentation. Instead of a straightforward phrase (e.g., "the trash can"), the query text may include more intricate expressions (e.g., "something that

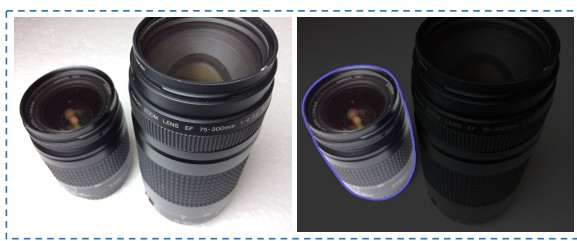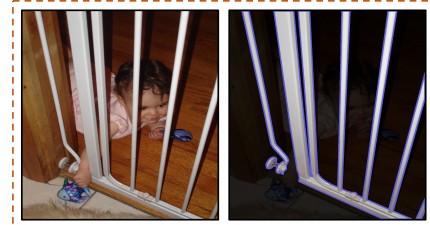

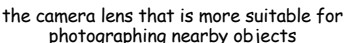

the camera lens that is more suitable for photographing nearby objects

Toddlers are curious and often enjoy exploring their surroundings. What object in the picture can provide a safe and enclosed space for a toddler to play in?

Figure 2: Examples of the annotated image-instruction pairs. Left: short query. Right: long query.

the garbage should be put into") or longer sentences (e.g., "After cooking, consuming food, and preparing for food, where can we throw away the rest of the food and scraps?") that involve complex reasoning or world knowledge.

## 3.2 BENCHMARK

Given the lack of quantitative evaluation, it is imperative to establish a benchmark for the reasoning segmentation task. To ensure reliable assessment, we have collected a diverse set of images from OpenImages (Kuznetsova et al., 2020) and ScanNetv2 (Dai et al., 2017), annotating them with implicit text instructions and high-quality target masks. To cover different scenarios, our text instructions consist of two types: 1) short phrases; 2) long sentences; as illustrated in Figure 2. The resulting *ReasonSeg* benchmark comprises a total of 1218 image-instruction pairs. This dataset is further partitioned into three splits: `train`, `val`, and `test`, containing 239, 200, and 779 image-instruction pairs, respectively. As the primary purpose of the benchmark is evaluation, the validation and testing sets include a larger number of image-instruction samples. The details of data annotation are given in Appendix A.2.

## 4 OUR METHOD

In this section, we first introduce the model architecture in Sec. 4.1. After that, we elaborate on the training data preparation and training parameters in Sec. 4.2.

### 4.1 ARCHITECTURE

**Embedding as Mask.** Most current multi-modal LLMs (such as LLaVA (Liu et al., 2023b), Flamingo (Alayrac et al., 2022), BLIP-2 (Li et al., 2023b), Otter (Li et al., 2023a), etc.) support image and text as input and text as output, but they cannot directly output fine-grained segmentation masks. VisionLLM (Wang et al., 2023) offers a solution by parsing segmentation masks as sequences of polygons, enabling the representation of segmentation masks as plain text and allowing end-to-end training within the framework of existing multi-modal LLMs. However, end-to-end training with the polygon sequences introduces optimization challenges and may compromise generalization ability unless a massive amount of data and computational resources are employed. For instance, training a 7B model, VisionLLM requires $4 \times 8$ NVIDIA 80G A100 GPUs and 50 epochs, which is computationally prohibitive. In contrast, training LISA-7B requires only 10,000 training steps on 8 NVIDIA 24G 3090 GPUs.

To this end, we propose the embedding-as-mask paradigm to infuse new segmentation capabilities into the multi-modal LLM. The pipeline of our method is illustrated in Fig. 3. Specifically, we first expand the original LLM vocabulary with a new token, i.e., <SEG>, which signifies the request for the segmentation output. Given a text instruction $\boldsymbol{x}_{txt}$ along with the input image $\boldsymbol{x}_{img}$, we feed them into the multi-modal LLM $\mathcal{F}$, which in turn outputs a text response $\hat{\boldsymbol{y}}_{txt}$. It can be formulated as

$$\hat{\boldsymbol{y}}_{txt} = \mathcal{F}(\boldsymbol{x}_{img}, \boldsymbol{x}_{txt}). \tag{1}$$

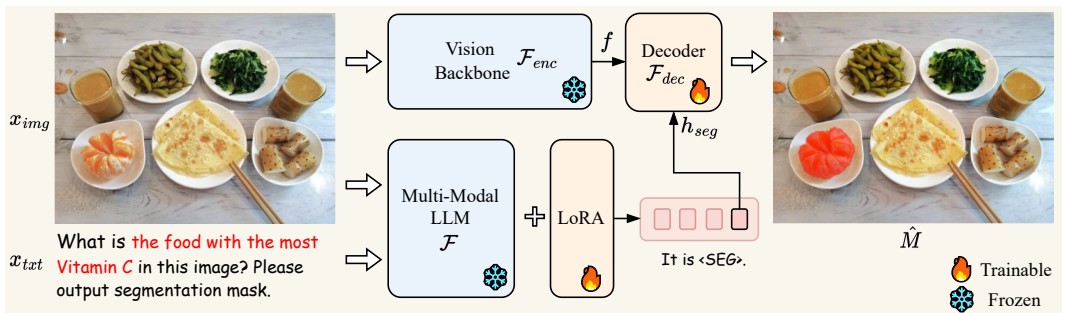

Figure 3: The pipeline of LISA. Given the input image and text query, the multi-modal LLM generates text output. The last-layer embedding for the `<SEG>` token is then decoded into the segmentation mask via the decoder. The choice of vision backbone can be flexible (e.g., SAM, Mask2Former).

When the LLM intends to generate a binary segmentation mask, the output $\hat{y}_{txt}$ should include a `<SEG>` token. We then extract the last-layer embedding $\hat{h}_{seg}$ corresponding to the `<SEG>` token and apply an MLP projection layer $\gamma$ to obtain $h_{seg}$. Simultaneously, the vision backbone $\mathcal{F}_{enc}$ extracts the dense visual embeddings $f$ from the visual input $x_{img}$. Finally, $h_{seg}$ and $f$ are fed to the decoder $\mathcal{F}_{dec}$ to produce the final segmentation mask $\hat{M}$. The detailed structure of the decoder $\mathcal{F}_{dec}$ follows Kirillov et al. (2023). The process can be formulated as

$$h_{seg} = \gamma(\hat{h}_{seg}), \quad f = \mathcal{F}_{enc}(x_{img}),$$
$$\hat{M} = \mathcal{F}_{dec}(h_{seg}, f). \tag{2}$$

**Training Objectives.** The model is trained end-to-end using the text generation loss $\mathcal{L}_{txt}$ and the segmentation mask loss $\mathcal{L}_{mask}$. The overall objective $\mathcal{L}$ is the weighted sum of these losses, determined by $\lambda_{txt}$ and $\lambda_{mask}$:

$$\mathcal{L} = \lambda_{txt}\mathcal{L}_{txt} + \lambda_{mask}\mathcal{L}_{mask}. \tag{3}$$

Specifically, $\mathcal{L}_{txt}$ is the auto-regressive cross-entropy loss for text generation, and $\mathcal{L}_{mask}$ is the mask loss, which encourages the model to produce high-quality segmentation results. To compute $\mathcal{L}_{mask}$, we employ a combination of per-pixel binary cross-entropy (BCE) loss and DICE loss, with corresponding loss weights $\lambda_{bce}$ and $\lambda_{dice}$. Given the ground-truth targets $y_{txt}$ and $M$, these losses can be formulated as

$$\mathcal{L}_{txt} = \mathbf{CE}(\hat{y}_{txt}, y_{txt}), \quad \mathcal{L}_{mask} = \lambda_{bce}\mathbf{BCE}(\hat{M}, M) + \lambda_{dice}\mathbf{DICE}(\hat{M}, M). \tag{4}$$

## 4.2 TRAINING

**Training Data Formulation.** As illustrated in Fig. 4, our training data comprises mainly three parts, all of which are derived from widely-used public datasets. The details are as follows:

- *Semantic Segmentation Dataset.* Semantic segmentation datasets typically consist of images and the corresponding multi-class labels. During training, we randomly choose several categories for each image. To generate data that matches the format of visual question answering, we employ a question-answer template like "**USER**: `<IMAGE>` `Can you segment the {class_name} in this image?` **ASSISTANT**: `It is <SEG>`.", where {class_name} is the chosen category, and `<IMAGE>` denotes the placeholder for tokens of image patches. The corresponding binary segmentation mask is used as the ground truth to provide mask loss supervision. During training, we also use other templates to generate the QA data to ensure data diversity, as shown in Appendix A.1. We adopt ADE20K, COCO-Stuff, and LVIS-PACO part segmentation datasets.

- *Vanilla Referring Segmentation Dataset.* Referring segmentation datasets provide an input image and an explicit short description of the target object. Thus, it is easy to convert them into question-answer pairs using a template like "**USER**: `<IMAGE>` `Can you segment {description} in this image?` **ASSISTANT**: `Sure, it is <SEG>`.", where {description} is the given explicit description. For this part, we adopt refCOCO, refCOCO+, refCOCOg, and refCLEF datasets.

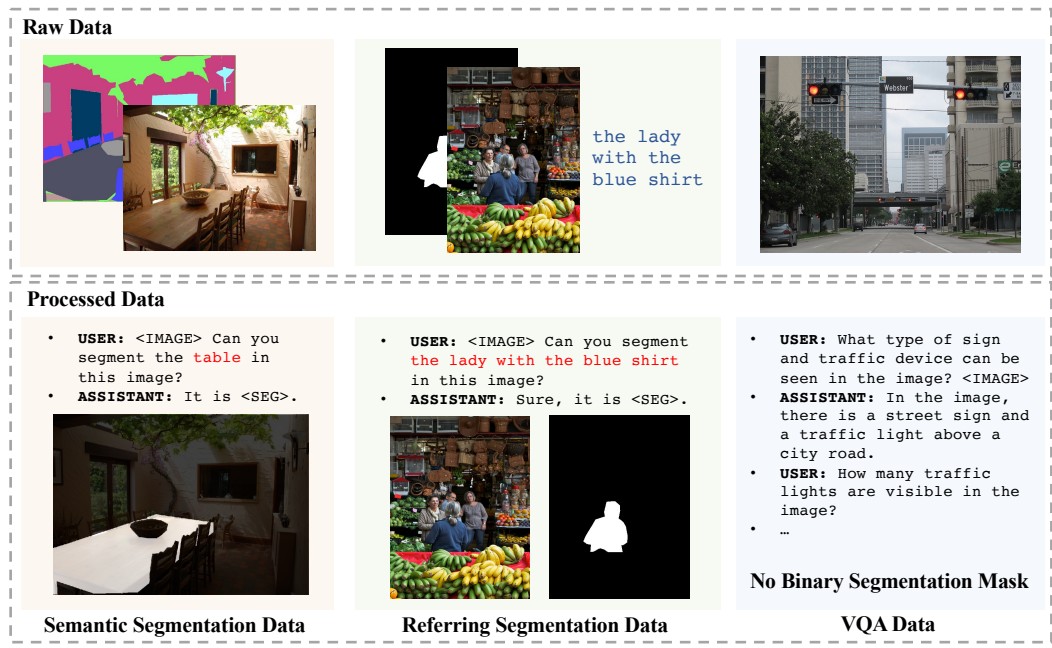

Figure 4: The illustration of training data formulation from different types of data, including semantic segmentation data, referring segmentation data, and visual question answering (VQA) data.

- *Visual Question Answering Dataset.* To preserve the original Visual Question Answering (VQA) ability of the multi-modal LLM (as shown in Appendix A.5), we also include the VQA dataset during training. We directly use the LLaVA-Instruct-150k data (Liu et al., 2023b) generated by GPT-4 (OpenAI, 2023).

Notably, the above datasets do not include any reasoning segmentation sample. Instead, it only contains samples where the target objects are explicitly indicated in the query texts. Surprisingly, even without complex reasoning training data, LISA demonstrates impressive zero-shot ability on the *ReasonSeg* benchmark, as shown in Table 1. Moreover, we find that further performance boost could be yielded by finetuning the model on only 239 image-instruction reasoning segmentation pairs.

**Trainable Parameters.** To preserve the generalization ability of the pre-trained multi-modal LLM $\mathcal{F}$ (i.e., LLaVA in our experiments), we leverage LoRA (Hu et al., 2021) to perform efficient fine-tuning, and completely freeze the vision backbone $\mathcal{F}_{enc}$. The decoder $\mathcal{F}_{dec}$ is fully fine-tuned. Additionally, the word embeddings of the LLM and the projection layer of $\gamma$ are also trainable.

## 5 EXPERIMENT

### 5.1 EXPERIMENTAL SETTING

**Network Architecture.** Unless otherwise specified, we use LLaVA-7B-v1-1 or LLaVA-13B-v1-1 as the multi-modal LLM $\mathcal{F}$, and adopt the ViT-H SAM (Kirillov et al., 2023) backbone as the vision backbone $\mathcal{F}_{enc}$. The projection layer of $\gamma$ is an MLP with channels of [256, 4096, 4096].

**Implementation Details.** We adopt 8 NVIDIA 24G 3090 GPUs for training. The training scripts are based on deepspeed (Rasley et al., 2020) engine. We use AdamW (Loshchilov & Hutter, 2017) optimizer with the learning rate and weight decay set to 0.0003 and 0, respectively. We also adopt WarmupDecayLR as the learning rate scheduler, where the warmup iterations are set to 100. The weights of the text generation loss $\lambda_{txt}$ and the mask loss $\lambda_{mask}$ are set to 1.0 and 1.0, respectively, and those of the bce loss $\lambda_{bce}$ and the dice loss $\lambda_{dice}$ are set to 2.0 and 0.5, respectively. Besides, the batch size per device is set to 2, and the gradient accumulation step is set to 10. During training, we select at most 3 categories for each image in semantic segmentation datasets.

Table 1: Reasoning segmentation results among LISA (ours) and previous related works. 'ft' denotes using 239 reasoning segmentation image-instruction pairs to finetune the model.

| Method | val | | test | | | | | | | |
|---|---|---|---|---|---|---|---|---|---|---|
| | overall | | short query | | long query | | overall | | | |
| | gIoU | cIoU | gIoU | cIoU | gIoU | cIoU | gIoU | cIoU | | |
| OVSeg (Liang et al., 2023) | 28.5 | 18.6 | 18.0 | 15.5 | 28.7 | 22.5 | 26.1 | 20.8 | | |
| GRES (Liu et al., 2023a) | 22.4 | 19.9 | 17.6 | 15.0 | 22.6 | 23.8 | 21.3 | 22.0 | | |
| X-Decoder (Zou et al., 2023a) | 22.6 | 17.9 | 20.4 | 11.6 | 22.2 | 17.5 | 21.7 | 16.3 | | |
| SEEM (Zou et al., 2023b) | 25.5 | 21.2 | 20.1 | 11.5 | 25.6 | 20.8 | 24.3 | 18.7 | | |
| LISA-7B | 44.4 | 46.0 | 37.6 | 34.4 | 36.6 | 34.7 | 36.8 | 34.1 | | |
| LISA-7B (ft) | 52.9 | 54.0 | 40.6 | 40.6 | 49.4 | 51.0 | 47.3 | 48.4 | | |
| LISA-13B | 48.9 | 46.9 | 39.9 | **43.3** | 46.4 | 46.5 | 44.8 | 45.8 | | |
| LISA-13B (ft) | **56.2** | **62.9** | **44.3** | 42.0 | **54.0** | **54.3** | **51.7** | 51.1 | | |

**Datasets.** As mentioned in Sec. 4.2, our training data is mainly composed of three types of datasets: (1) For the semantic segmentation dataset, we use ADE20K (Zhou et al., 2017) and COCO-Stuff (Caesar et al., 2018). Besides, to enhance the segmentation results for some part of an object, we also use part semantic segmentation datasets, including PACO-LVIS (Ramanathan et al., 2023), PartImageNet (He et al., 2022), and PASCAL-Part (Chen et al., 2014); (2) For the referring segmentation dataset, we use refCLEF, refCOCO, refCOCO+ (Kazemzadeh et al., 2014), and refCOCOg (Mao et al., 2016). (3) For the visual question answering (VQA) dataset, we use LLaVA-Instruct-150k dataset (Liu et al., 2023b). In order to avoid data leakage, we exclude the COCO samples whose images are present in the refCOCO(+/g) validation sets during training. Furthermore, we surprisingly find that by finetuning the model on only 239 samples of ReasonSeg image-instruction pairs, the model's performance can be further boosted.

**Evaluation Metrics.** We follow most previous works on referring segmentation (Kazemzadeh et al., 2014; Mao et al., 2016) to adopt two metrics: gIoU and cIoU. gIoU is defined by the average of all per-image Intersection-over-Unions (IoUs), while cIoU is defined by the cumulative intersection over the cumulative union. Since cIoU is highly biased toward large-area objects and it fluctuates too much, gIoU is preferred.

## 5.2    REASONING SEGMENTATION RESULTS

The reasoning segmentation results are shown in Table 1. It is worth noting that existing works fail to handle the task, but our model can accomplish the task involving complex reasoning with more than 20% gIoU performance boost. As mentioned before, the reasoning segmentation task is essentially different from the previous referring segmentation task in that it requires the model to possess *reasoning ability* or access *world knowledge*. Only by truly understanding the query, can the model do well in the task. The existing works are limited to explicit referring and have no proper way to understand an implicit query, but our model exploits multi-modal LLMs to reach the goal.

Another finding is that LISA-13B outperforms the 7B counterpart substantially, especially on the long-query scenarios, which indicates that the current performance bottleneck may still lie in understanding the query text, and a stronger multi-modal LLM might lead to even better results.

## 5.3    VANILLA REFERRING SEGMENTATION RESULTS

To show that our model is also competent in the vanilla referring segmentation task, we make a comparison with existing state-of-the-art methods in Table 2. We evaluate the methods on refCOCO, refCOCO+, refCOCOg validation and testing sets. Our model achieves state-of-the-art results across various referring segmentation benchmarks.

## 5.4    ABLATION STUDY

In this section, we conduct an extensive ablation study to reveal the contribution of each component. Unless otherwise specified, we report the metrics of gIoU and cIoU of LISA-7B on the validation set.

Table 2: Referring segmentation results (cIoU) among LISA (ours) and existing methods. 'ft' denotes using the referring segmentation datasets (refCOCO(+/g)) to finetune the model.

| Method | refCOCO | | | refCOCO+ | | | refCOCOg | |
|---|---|---|---|---|---|---|---|---|
| | val | testA | testB | val | testA | testB | val(U) | test(U) |
| MCN (Luo et al., 2020) | 62.4 | 64.2 | 59.7 | 50.6 | 55.0 | 44.7 | 49.2 | 49.4 |
| VLT (Ding et al., 2021) | 67.5 | 70.5 | 65.2 | 56.3 | 61.0 | 50.1 | 55.0 | 57.7 |
| CRIS (Wang et al., 2022) | 70.5 | 73.2 | 66.1 | 62.3 | 68.1 | 53.7 | 59.9 | 60.4 |
| LAVT (Yang et al., 2022) | 72.7 | 75.8 | 68.8 | 62.1 | 68.4 | 55.1 | 61.2 | 62.1 |
| ReLA (Liu et al., 2023a) | 73.8 | 76.5 | 70.2 | **66.0** | **71.0** | 57.7 | 65.0 | 66.0 |
| X-Decoder (Zou et al., 2023a) | - | - | - | - | - | - | 64.6 | - |
| SEEM (Zou et al., 2023b) | - | - | - | - | - | - | 65.7 | - |
| LISA-7B | 74.1 | 76.5 | 71.1 | 62.4 | 67.4 | 56.5 | 66.4 | 68.5 |
| LISA-7B (ft) | **74.9** | **79.1** | **72.3** | 65.1 | 70.8 | **58.1** | **67.9** | **70.6** |

Table 3: Ablation study on the design choice of vision backbone. 'ft' denotes fine-tuning on ReasonSeg training set.

| Vision Backbone | gIoU | cIoU |
|---|---|---|
| Mask2Former-Swin-L | 42.4 | 38.8 |
| SAM (w/ LoRA) | 41.5 | 37.3 |
| SAM | 44.4 | 46.0 |
| Mask2Former-Swin-L (ft) | 50.7 | 52.3 |
| SAM w/ LORA (ft) | 51.8 | 51.9 |
| SAM (ft) | **52.9** | **54.0** |

Table 4: Ablation study on SAM pre-trained weight, MLP for projection layer $\gamma$, and rephrasing.

| Exp. ID | Pre-train$_{SAM}$ | MLP $\gamma$ | rephrasing | gIoU | cIoU |
|---|---|---|---|---|---|
| 1 | | ✓ | ✓ | 35.9 | 44.6 |
| 2 | ✓ | | ✓ | **53.2** | 51.0 |
| 3 | ✓ | ✓ | | 50.7 | 51.1 |
| 4 | ✓ | ✓ | ✓ | 52.9 | **54.0** |

Table 5: Ablation study on training data.

| Exp. ID | SemanticSeg | | | ReferSeg | VQA | ReasonSeg | gIoU | cIoU |
|---|---|---|---|---|---|---|---|---|
| | ADE20K | COCO-Stuff | PartSeg | | | | | |
| 1 | | ✓ | ✓ | ✓ | ✓ | ✓ | 48.9 | 53.5 |
| 2 | ✓ | | ✓ | ✓ | ✓ | ✓ | 48.5 | 50.8 |
| 3 | ✓ | ✓ | | ✓ | ✓ | ✓ | 46.7 | 50.9 |
| 4 | | | ✓ | ✓ | ✓ | ✓ | 46.6 | 46.7 |
| 5 | | | | ✓ | ✓ | ✓ | 30.4 | 20.4 |
| 6 | ✓ | ✓ | ✓ | | ✓ | ✓ | 47.7 | 51.1 |
| 7 | ✓ | ✓ | ✓ | ✓ | ✓ | | 44.4 | 46.0 |
| 8 | ✓ | ✓ | ✓ | ✓ | ✓ | ✓ | **52.9** | **54.0** |

**Design Choices of Vision Backbone.** We emphasize that vision backbones other than SAM are also applicable in our framework. To verify this fact, we conduct ablation in Table 3. No matter whether we finetune the model on ReasonSeg training set, SAM performs better than Mask2Former-Swin-L. We explain that SAM is trained with billions of high-quality masks, and thus yields a higher metric than Mask2Former that is trained on merely the COCO dataset (Lin et al., 2014). We also notice that even with Mask2Former, our framework achieves a decent performance on the reasoning segmentation task, significantly outperforming previous works such as X-Decoder (Zou et al., 2023a). This reveals the fact that the design choice of vision backbone is flexible and not limited to SAM.

**SAM LoRA Fintuning.** We also investigate the effectiveness of applying LoRA on SAM backbone. In Table 3, we note that the performance of LoRA finetuned SAM backbone is inferior to that of the frozen one. A potential reason is that fine-tuning impairs the generalization ability of the origianl SAM model.

**SAM Pre-trained Weight.** To demonstrate the contribution of SAM pre-trained weight, we make a comparison between Experiments 1 and 4 in Table 4. Without being initialized by SAM pre-trained

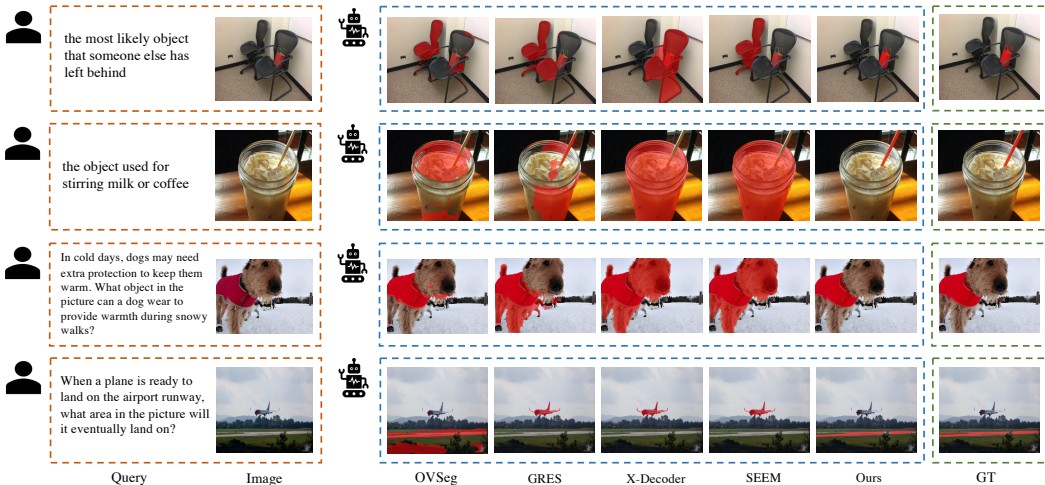

Figure 5: Visual comparison among LISA (ours) and existing related methods. More illustrations are given in Appendix A.4.

weight, the vision backbone is trained from scratch. This causes the performance falling behind that of the baseline model substantially.

**MLP vs. Linear Projection Layer.** In experiments 2 and 4 of Table 4, we notice that making $\gamma$ an MLP yields little performance decrease in gIoU, but a relatively higher performance in cIoU.

**Contribution of All Types of Training Data.** In Table 5, we show the contribution of each type of data to the performance. It is worth noting that in Exp. 4, we do not use any semantic segmentation dataset, and the performance drops a lot. We conjecture that semantic segmentation datasets provides a large amount of ground-truth binary masks for training, since a multi-class label can induce multiple binary masks. This shows that semantic segmentation datasets are crucial in training.

**Instruction Rephrasing by GPT-3.5.** During finetuning on the reasoning segmentation image-instruction pairs, we rephrase the text instruction by GPT-3.5 (as shown in Appendix A.3), and randomly choose one. The comparison between Experiments 3 and 4 in Table 4 shows that the performance is increased by 2.2% gIoU and 2.9% cIoU. This result verifies the effectiveness of such data augmentation.

## 5.5 QUALITATIVE RESULTS

As depicted in Fig. 5, we provide a visual comparison with existing related works, including the model for open-vocabulary semantic segmentation (OVSeg), referring segmentation (GRES), and the generalist models for segmentation (X-Decoder and SEEM). These models fail to handle the displayed cases with various errors, while our approach produces accurate and high-quality segmentation results. More illustrations are given in Appendix A.4.

## 6 CONCLUSION

In this work, we have proposed a new segmentation task—*reasoning segmentation*. This task is significantly more challenging than the vanilla referring segmentation task, as it requires the model to actively reason based on implicit user instructions. To enable effective evaluation, we have introduced a benchmark for this task, namely *ReasonSeg*. We hope this benchmark will be beneficial for the development of related technologies. Finally, we have presented our model — LISA. By employing the embedding-as-mask paradigm, it injects new segmentation capabilities into current multi-modal LLMs and performs surprisingly well on the reasoning segmentation task, even when trained on reasoning-free datasets. Consequently, it demonstrates the ability to chat with segmentation mask outputs in various scenarios. We believe our work will shed new light on the direction of combining LLMs and vision-centric tasks.

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

# A APPENDIX

## A.1 TEMPLATES USED IN TRAINING DATA FORMULATION

When formulating the training data, we convert the raw data into the question-answer-mask triples with the following templates:

---

For short-phrase descriptions, we use the following templates:

- `<IMAGE> Can you segment the {short_phrase} in this image?`

- `<IMAGE> Please segment the {short_phrase} in this image.`

- `<IMAGE> What is {short_phrase} in this image? Please respond with segmentation mask.`

- `<IMAGE> What is {short_phrase} in this image? Please output segmentation mask.`

---

For long-sentence descriptions, we use the following templates:

- `<IMAGE> {long_sentence} Please respond with segmentation mask.`

- `<IMAGE> {long_sentence} Please output segmentation mask.`

---

For explanatory descriptions, we use the following templates:

- `<IMAGE> Please output segmentation mask and explain why.`

- `<IMAGE> Please output segmentation mask and explain the reason.`

- `<IMAGE> Please output segmentation mask and give some explanation.`

---

For answers, we use the following templates:

- `It is <SEG>.`

- `Sure, <SEG>.`

- `Sure, it is <SEG>.`

- `Sure, the segmentation result is <SEG>.`

- `<SEG>.`

---

## A.2 DATA ANNOTATION OF REASONSEG

Each image in the proposed ReasonSeg benchmark is accompanied by a reasoning query and a binary mask to identify the target region or objects. However, acquiring the query may not always be straightforward and can significantly contribute to the overall time cost of the process. Therefore, we leverage a semi-automatic process that supplements full human annotation to facilitate the annotation process for reasoning segmentation.

Specifically, we first manually annotate the query and binary mask for nearly 300 samples. After that, we utilize existing tools to assist in the annotation process with the following steps:

- Step-1: Each image is processed using RAM (Zhang et al., 2023b) to extract descriptive tags for the objects and elements present in the image.

- Step-2: The extracted tags are incorporated as part of the text prompt provided to GPT-3.5, generating five alternative queries. Additionally, the human annotations are also included as part of the text prompt.

- Step-3: Human experts evaluate and select the most suitable query from the five options. In cases where none of the options are satisfactory, human experts may manually annotate the sample.

The prompt construction process adopted in Step-2 is presented in Table 6, and some examples used for prompting are shown in Table 7.

Table 6: The illustration of the prompt construction process for generating alternative queries.

```
messages = ["role":"system", "content": You are an AI visual assistant, and you are
seeing a single image. What you see are provided with several words, describing the same image you
are looking at. Design a question you may have when you look at this image, based on the description
words. The questions should be in a tone that a visual AI assistant is seeing the image, asking diverse
questions and give corresponding answers related to the image content. Include questions asking
about the visual content of the image, including the objects or the object parts considering the object
characteristics, the relations between objects, etc. Only include questions that have definite answers:
(1) one can see the content in the image that the <question> asks about and can answer confidently
with the <answer>; (2) one can determine confidently from the image that it is not in the image. Do
not ask any <question> that cannot be answered confidently. (3) each question is only allowed to
have one answer that corresponds to it the most, and there should not be multiple possible answers in
the list of description words. (4) the answer must be nouns related to actual objects or items, instead
of color, environment, action, texture, etc. Please include complex questions that are relevant to the
content in the image, for example, asking about background knowledge of the objects in the image,
asking to discuss events related to the objects in the image, etc. Also, do not ask about uncertain
details. ]
for sample in fewshot_samples:
    messages.append({"role":"user", "content":sample["context"]})
    messages.append({"role":"assistant", "content":sample["response"]}
    )
messages.append({"role":"user", "content":'\n'.join(query)})
```

Table 7: Illustrations of the examples used for prompting.

---

"context": antler, trumpet, dog, hat, head, husky, leash, moose, red, reindeer, stare, wear
"response": <question> Dogs do not have horns on their heads, only a pair of ears. What part of the dog's head in this picture looks strange?
<answer> antler.

---

"context": alcohol, beer, beverage, coffee, coffee cup, spoon, cup, table, dinning table, plate, drink, food, glass table, juice, liquid, platter, saucer, silverware, tea, tea pot, tray, utensil
"response": <question> Pure black coffee can be too bitter to drink, so people often add sugar to it to sweeten the taste. What tool in this image can be used to add sugar to coffee and stir it?
<answer> spoon.

---

"context": adder, binocular, pole, stand, tree
"response": <question> It can be difficult for people to climb up a bare pole and inspect or repair the upper part. What object is the person in the picture relying on to accomplish this task?
<answer> ladder.

---

"context": cave, cliff, pillar, formation, lagoon, lake, pole, stone, rock face, rock formation, stand, tree trunk, water
"response": <question> If we were at the location shown in the picture and did not consider diving underwater, what area in the picture could we explore further?
<answer> cave.

---

"context": apron, broom, catch, cloak, costume, doll, dress, ghost, hair, Halloween, man, puppet, pitchfork, rake, robe, strawman, skeleton, wear, wig, witch
"response": <question> If the character in the picture is a wizard from the Harry Potter world, what would he ride on to fly in the air?
<answer> broom.

---

"context": boat, paddle, rowboat, dragon, drummer, canoe, person, life jacket, man, oar, ride, row, vessel, water
"response": <question> In an intense dragon boat race. What should be struck to boost the morale of the competing team and cheer them on?
<answer> drum.

---

"context": black, cocker, dog, floor, grass, green, hang, lay, lush, mouth, neckband, toy, ball
"response": <question> Dogs are faithful companions to humans, and humans often play fetch games with them. What object will the dog likely retrieve and bring back to the human for the next round of fetch in the picture?
<answer> ball.

---

"context": armchair, bedroom, bookshelf, lamp, bureau, carpet, ceiling fan, chair, computer, computer desk, couch, den, table, drawer, dresser, floor, furniture, living room, office chair, office supply, room, stool, swivel chair
"response": <question> Suppose you are a smart home AI. It's very hot outside, and the owner comes back home. What appliance would you turn on to help cool down the temperature?
<answer> ceiling fan.

---

## A.3 INSTRUCTION REPHRASING

We adopt GPT-3.5 to rephrase the instruction queries for samples in ReasonSeg. The system prompt and examples are as follows.

Table 8: Illustrations of the system message and examples for instruction rephrasing.

```
messages = ["role":"system", "content": You are an AI language assistant, and your
role is to assist users in rephrasing their questions to new formats while preserving the same answer. It
is important to ensure that the same answer can be used to respond to both the original and rephrased
versions of the question. You may refer to the provided examples for guidance, and they may represent
different ways of rephrasing. Please follow these examples to finish rephrasing. ]
for sample in fewshot_samples:
    messages.append({"role":"user", "content":sample["context"]})
    messages.append({"role":"assistant", "content":sample["response"]}
    )
messages.append({"role":"user", "content":'\n'.join(query)})
```

"context": Industrial activities may produce a large amount of waste gas, which can cause air pollution. What part of the picture indicates the presence of waste gas being released?
"response": Please locate the part of the picture that indicates the presence of waste gas being released.

"context": Industrial activities may produce a large amount of waste gas, which can cause air pollution. What part of the picture indicates the presence of waste gas being released?
"response": Please locate the part of the picture that indicates the presence of waste gas being released.

"context": I feel my commute is too slow now and I'm hoping to find a convenient mode of transportation that can also help me exercise. Can you help me find the corresponding part in the picture?
"response": Could you point me to the part of the picture that showcases a convenient mode of transportation that can also provide exercise, as you're seeking to improve the efficiency of the commute?

"context": Pure black coffee can be too bitter to drink, so people often add sugar to it to sweeten the taste. What tool in this image can be used to add sugar to coffee and stir it?
"response": The item that can be used to add sugar to coffee and stir it simultaneously.

"context": We cannot breathe underwater, so diving requires additional equipment to help people breathe while underwater. What in the picture can help humans accomplish this task?
"response": Since humans can't breathe underwater, diving typically involves additional equipment to enable breathing while submerged. Based on this image, which item could be utilized to assist with this task?

"context": What object can provide light for a room?
"response": As an intelligent robot, what appliance would you turn off when the owner says they want to go to sleep to make the room darker and more suitable for sleeping?

## A.4 More Qualitative Results

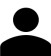 Can you segment the camera lens that is more suitable for photographing nearby objects in this image?

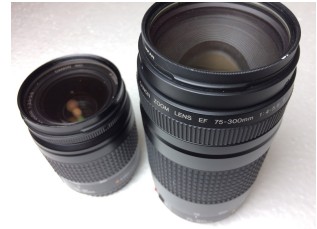

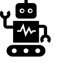 Sure, [SEG].

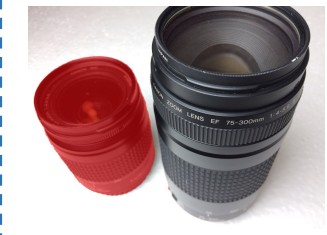

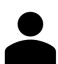 Who was the president of the US in this image? Please output segmentation mask and explain the reason.

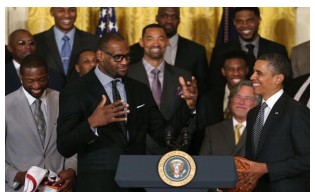

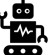 Sure, the segmentation result is [SEG]. The President of the United States in the image is President Obama.

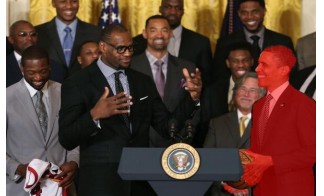

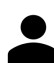 Who was the president of the US in this image? Please output segmentation mask and explain why.

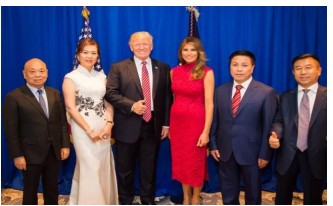

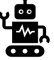 Sure, [SEG]. In the image, the President of the United States is President Trump.

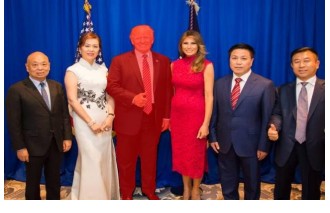

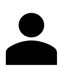 Can you segment the founder of Alibaba in this image and explain why?

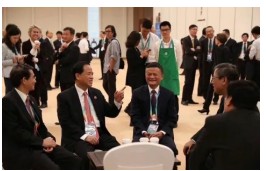

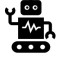 Sure, it is [SEG]. In the image, the man wearing a red tie and sitting on one of the chairs is Jack Ma, the co-founder of Alibaba Group...

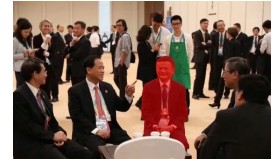

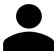 What is something that prevents people from getting into the building in this image? Please output segmentation mask.

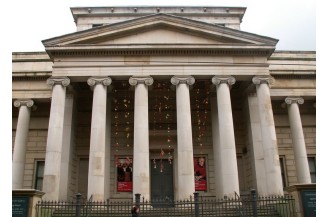

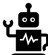 <SEG>.

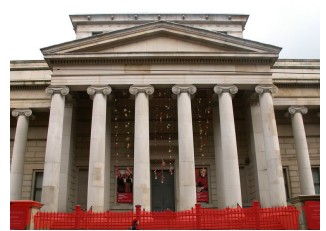

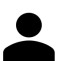 Where can the driver see the car speed in this image? Please output segmentation mask and explain why.

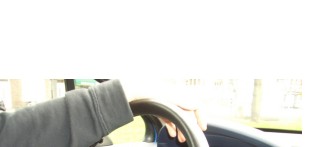

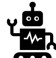 Sure, it is <SEG>. The driver can see the car speed on the speedometer, which is located in front of them while sitting in the driver's seat.

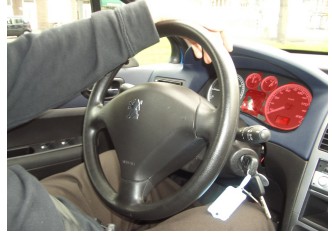

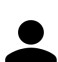 What can make the woman stand higher? Please output segmentation mask and explain why.

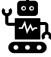 Sure, <SEG>. The woman is standing higher by using a ladder to reach the top of the house. She is holding onto the ladder for support and stability while working on the roof. This allows her to access areas that would be difficult or unsafe to reach from the ground level.

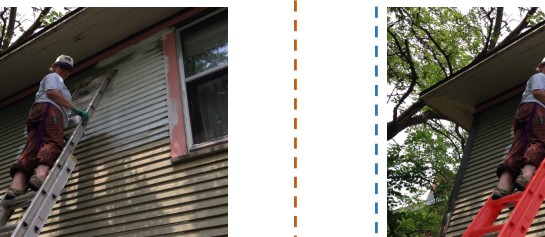

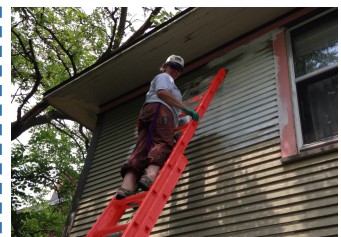

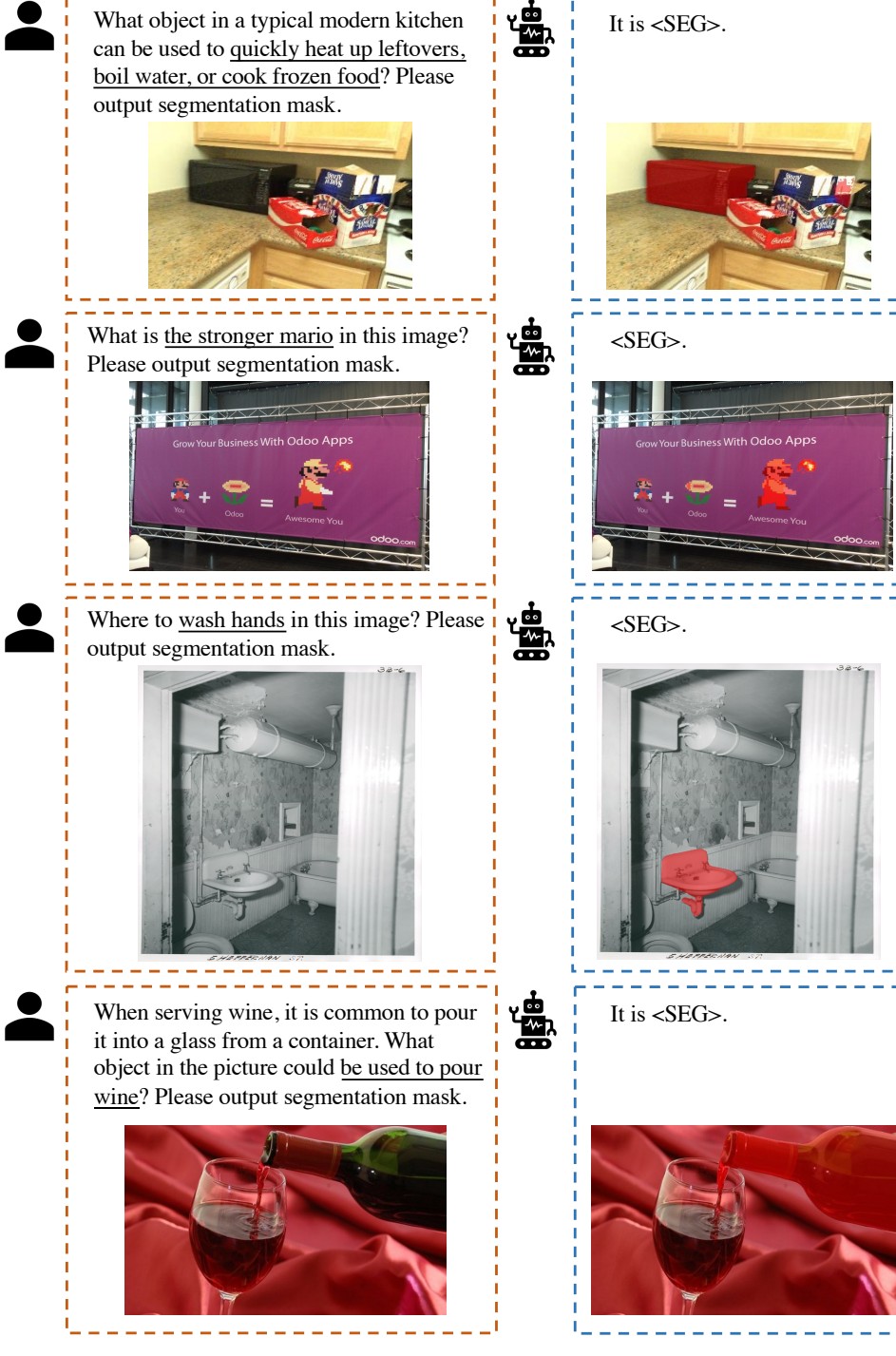

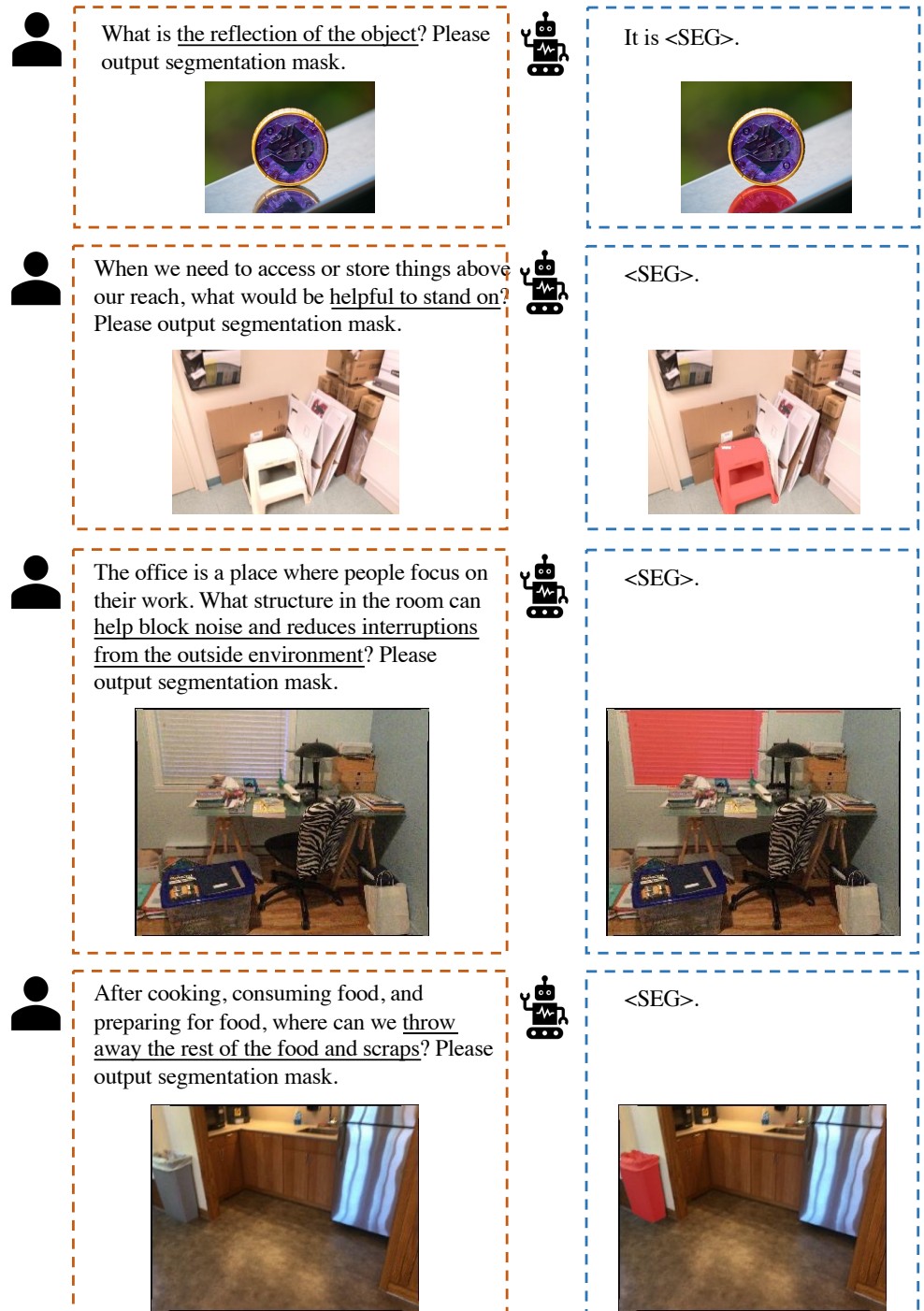

Figure 6: More qualitative results.

## A.5 Conversation Ability of LISA

We emphasize that our model — LISA not only fulfills the reasoning segmentation capability but also inherits the conversation ability from current multi-modal LLMs. Our model does not lose the capability of text generation, potentially because (1) a VQA dataset (i.e., LLaVA-Instruct-150k) is incorporated for training, and (2) only a small part of model parameters are trainable. We demonstrate the Visual Question Answering (VQA) ability in Fig. 7 below.

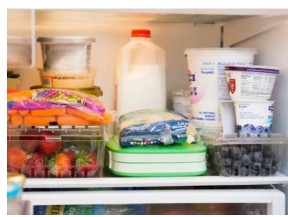 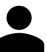

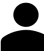
What are the meals that I can cook with these?

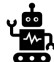
With the variety of food items in the refrigerator, you can create a range of meals. For instance, you can make a salad using the lettuce, carrots, and strawberries. The carrots can also be used to make a vegetable soup or stew, while the lettuce can be utilized in a fresh salad or as a base for a sandwich. The milk can be used to make creamy dishes, such as pasta or risottos, or to add to your coffee or tea. Additionally, the yogurt can be incorporated into smoothies, parfaits, or as a topping for oatmeal or granola. The strawberries can be used to make a fruit salad, a strawberry shortcake, or a strawberry yogurt parfait. The variety of options is limited only by your imagination and the dishes you prefer to prepare.

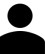
I would like to do the fruit salad. Show me the recipe.

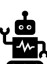
Sure, the fruit salad is a simple and healthy option that can be easily made using the ingredients in the refrigerator. Here is a quick recipe for a strawberry and carrot fruit salad:
Ingredients:
- 1 cup fresh strawberries, hulled and sliced
- 1 cup fresh carrots, peeled and grated
- 1/2 cup green grapes, halved
- 1/4 cup chopped walnuts ( option)
- 1/4 cup honey (adjust to taste)
- 2 tablespoons lemon juice
Instructions:
1. In a large bowl, combine the sliced strawberries, grated carrots, and halved grapes.
2. In a small bowl, whisk the honey and lemon juice until well combined.
3. pour the honey-ler juice mixture over the fruit and toss gently to coat.
4. If desired, add the chopped walnuts and toss gently to mix.
5. chill the fruit salad in the refrigerator for at least 30 minutes before serving.
This fruit salad is a colorful and refreshing option, perfect for a light lunch or a healthy snack. The combination of sweet and tangy flavors, along with the crunch from the walnuts, makes it a delightful and satisfying dish.

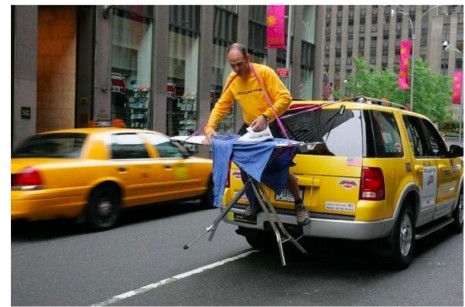 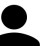

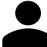

What is unusual about this image? 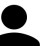

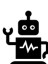 The unusual aspect of this image is that a man is standing on top of a yellow car, which is parked on the side of a road. This is not a common sight, as people typically do not stand on top of moving vehicles or parked cars, especially in a busy city street with traffic. The man is also holding a ironing board, which is an unconventional and unexpected item to be seen in such a setting. This scene is likely staged for a humorous or artistic purpose, as it is not a typical scenario that one would encounter in everyday life.

[Start a new conversation, and clear the history]

What's happening in the scene? 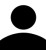

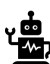 The image depicts a man hanging his clothes on the back of a yellow van while it's parked on the side of a road. The van is surrounded by traffic, with several cars and a taxi passing by in the background. The van is parked near a sidewalk, and the man is standing on the back of the van, holding his ironing board.There are multiple people in the scene, with one person standing near the van, another person standing further away, and a third person standing near the edge of the image. The scene captures the busy nature of the street and the van owner's unique way of utilizing their vehicle.

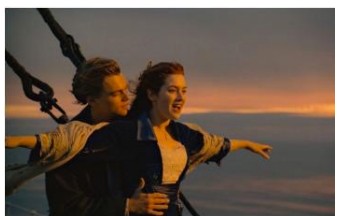

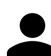

What's the ending of this movie?

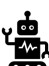
The ending of the movie "Titanic" shows the main characters, Jack and Rose, standing on the bow of the ship with their arms outstretched, flying through the air as the ship sinks. This iconic scene has become a symbolic representation of the tragedy that occurred when the real-life Titanic sank on its maiden voyage in 1912. The movie's climax is a powerful and emotional moment that captures the essence of the disaster and the love story between Jack and Rose.

[Start a new conversation, and clear the history]

Sometimes I just look at pictures of the earth from space and I marvel at how beautiful it all is.

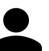

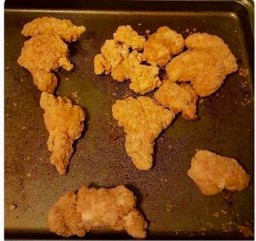

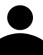
Can you explain this meme in detail?

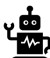
The image is a playful meme that features a tray of fried chicken with various countries cut out of the chicken patties. The countries are shaped from the chicken patties and placed on a baking tray. The countries include Africa, Europe, and the United States.The meme is a creative and fun way to showcase the countries on a tray of fried chicken, making it an interesting and visually appealing representation of geography. It demonstrates how food can be used as a tool for teaching or learning about different countries and their shapes.

Figure 7: Illustrations of Visual Question Answering (VQA) task. They show that LISA still maintains the capabilities of conversations.

