# OpenReview forum: "LISA: Reasoning Segmentation via Large Language Model"
_ICLR.cc/2024/Conference — ICLR 2024 Conference Withdrawn Submission_

### Official Review · Reviewer_VGzJ · 2023-10-24

**Soundness:** 3 good
**Presentation:** 3 good
**Contribution:** 3 good
**Rating:** 6
**Confidence:** 4

**Summary:**

This paper presented LISA, a framework for language-instructed segmentation that combines the language generation and segmentation ability from SAM. This is achieved mainly by introducing the special token <SEG>. They also collect and construct relevant datasets and benchmarks.

**Strengths:**

- A novel and interesting task setting: reasoning segmentation task.
- The authors used a special token <SEG> to bridge the LLM and SAM, and used the LoRA technique to fine-tune the model. In this way, the LISA could contain both LLM's reasoning ability and the SAM's segmentation ability.
- The usage of LoRA could help the model to achieve good performance with a limited training dataset and computation source, which brings light for most labs in the community.
- Extensive experiments and good performance.
- The authors gave details wrt the implementation details, especially the promotes used to diversity the datasets, which brings some insightful knowledge to the community.

**Weaknesses:**

- More experiments are needed to make the claim clear: the authors mention that "239 reasoning segmentation image-instruction pairs results in further performance enhancement", this is impressive. However, where comes the number? Would the performance get further boosted with more data samples?

- Some direct while simple comparison is needed: how about just using Grounding-SAM, the combination of Grounding-DINO with SAM? What would be the strengths of the LISA.

- Some writing is unclear, for example, I noticed many tables contain the items with "(ft)" while they seem to have different meanings.

- Currently, only one segmentation instance can be obtained during inference. Also, it is reasonable due to the limited dataset source, could the author demonstrate the potential scalability.

**Questions:**

- See Weakness.

- I am curious about the training cost, the author only mentions the training source is "8 NVIDIA 24G 3090 GPUs for training" how long would it take?

- In table 4, "MLP for projection layer γ" seems only harm the performance, can the authors give some insights wrt this? If not used, how to align the feature channels?

- Will more dataset sample, rather than 239 samples help the final results?

- Could the proposed methods be easily adapted to other domains, like remote sensing, etc?

- Since the QA pair is not that hard, I am wondering would the classic T5 / Bert achieve a similar performance?

- I am afraid that the claims "Remarkably, LISA can handle cases involving: 1) complex reasoning; 2) world knowledge; 3) explanatory answers; 4) multi-turn conversation." need more experiments to support them. Especially, the "multi-turn conversation" is only demonstrated with pure-text.

---

### Official Review · Reviewer_1t4G · 2023-10-28

**Soundness:** 3 good
**Presentation:** 2 fair
**Contribution:** 2 fair
**Rating:** 5
**Confidence:** 4

**Summary:**

This paper introduces a new segmentation task, named Reasoning Segmentation (ReasonSeg). Similar to the existing task "Referring Expression Segmentation" (RES), the new task requires a model to take in a free-form natural language description and a reference image, then output a desirable segmentation mask as required by the language input. The main difference between the two is that the language input in ReasonSeg is longer and less restrictive compared to RES. The authors also proposed LISA, leveraging existing multimodal LLMs with added trainable layers and special token <SEG> to achieve the ability required by Reasoning Segmentation. LISA is trained on multiple datasets adapted from Semantic Segmentations, VQA, RES tasks, and a new set of annotated image-instruction reasoning segmentation pairs. The proposed model LISA performes well on ReasonSeg as well as RES.

**Strengths:**

1. The new task Reasoning Segmentation (ReasonSeg) seems like a natural progression from its similar task: Referring Expression Segmentation (RES) and Open-Vocabulary Segmentation (OVSeg). ReasonSeg is more challenging, less restrictive, and more flexible in real-world applications.
2. The proposed pipeline LISA is a straightforward and effective solution in adapting existing models and datasets to achieve the challenging ReasonSeg task with reasonable training requirements.
3. This paper yields quite some empirical insights which all could be useful to the research community.

**Weaknesses:**

1. LISA has limited technical contributions in its design.
2. Although the new task ReasonSeg is well motivated, (which requires two key capabilities: 1. long text understanding and 2. segmentation), LISA's capabilities are not fully motivated and evaluated. As described by the author: "LISA can handle various scenarios, including 1) complex reasoning; 2) world knowledge; 3) explanatory answers; and 4) multi-turn conversations.", however, this paper mainly evaluates LISA's capability in segmentation with complex reasoning via ReasonSeg and RES benchmarks. Capabilities 2,3,4 (especially 3,4) on LISA are not required by the task ReasonSeg. Would a model similar to LISA but with segmentation output only perform better?
3. Referring to the templates used (Section A. 1) in adapting the existing datasets, if the goal is to preserve the conversational ability (which is not motivated well), why not use LLMs to rephrase these templates? If the goal is to achieve ReasonSeg only, why not output the <SEG> only?
4. The scale of the new dataset ReasonSeg is relatively small. It would be much more helpful to the community if the dataset is bigger, and the following works can keep on researching the 4 capabilities mentioned but not quite investigated.

**Questions:**

Please refer to the weaknesses.

---

### Official Review · Reviewer_KBow · 2023-10-29

**Soundness:** 3 good
**Presentation:** 4 excellent
**Contribution:** 3 good
**Rating:** 5
**Confidence:** 4

**Summary:**

This paper proposes a new task termed reasoning-based segmentation, and designs a simple yet effective method that integrates a pretrained vision expert, such as SAM, into the LLM by enabling it to accept the LLM's output embedding as the input. The method shows promising performance that can interpret abstract human instructions to segment the desired objects.

**Strengths:**

This paper proposes an interesting task, and the proposed method seems effective and promising. The utilization of pretrained vision expert seems to be a clever way of enabling vision ability of the LLM.

**Weaknesses:**

I have the following concerns about the paper:
1. I wonder if the model is able to perform instance segmentation, is it able to output multiple masks in one answer? For example, if there are two men, can I obtain answer like: the mask for the first man <seg>, and the mask for the second man <seg> ?
2. I wonder how the model performs on text-generation task, does the model preserve the original ability to perform conversation? I hope the authors can experimentally verify this.

I will consider updating my score, depending on the author's response.

**Questions:**

See weakness.

---

### Official Review · Reviewer_t6An · 2023-11-01

**Soundness:** 3 good
**Presentation:** 3 good
**Contribution:** 2 fair
**Rating:** 5
**Confidence:** 4

**Summary:**

This work addresses the limitations of existing perception systems, which rely on explicit human instructions for visual recognition tasks, often failing to comprehend implicit user intentions. The authors introduce a novel segmentation task called "reasoning segmentation," where complex and implicit text queries are used to generate segmentation masks for given images. They establish a benchmark with over a thousand image-instruction pairs that require intricate reasoning and world knowledge.

The authors present LISA (Large Language Instructed Segmentation Assistant), a multi-modal Large Language Model (LLM) capable of producing segmentation masks. LISA extends the vocabulary with a <SEG> token and utilizes an "embedding-as-mask" paradigm to enable segmentation capabilities. LISA handles scenarios involving complex reasoning, world knowledge, explanatory answers, and multi-turn conversations. It also demonstrates robust zero-shot capability when trained on reasoning-free datasets.

In summary, this work offers a model that can comprehend complex and implicit queries to generate segmentation masks effectively. It not only generates segmentation for language description but also performs well in multiple types of segmentation tasks.

**Strengths:**

The paper proposed an interesting view to generate segmentations from language inputs. It utilizes LLMs and multimodal LLMs to understand language sentences as input and produce segmentation embedding as output.
The experiments evaluate the model performances from multiple segmentations which demonstrated the improvements.

**Weaknesses:**

The work uses pre-trained LLMs and MLLMs as pre-stage, so it involves more learned external knowledge in the proposed pipeline. This would not be fair enough for the methods without using pre-trained foundation models.

The token "SEG" is only one token designed for the task, so how to use it for multiple segmentation masks

The reasoning question can be recognized by foundation models, so the reasoning capacity of the model actually not from the proposed components.

**Questions:**

How to fuse the h_seg and f_dec? Which way is the best? Thsese should be explained if you have more experiments.